

# Scenario-based impacts of land use and climate changes on the hydrology of a lowland rainforest catchment in Ghana, West Africa

Michael S. Aduah[1], Graham. P.W. Jewitt[2], Michele L.W. Toucher[2, 3]

[1]Department of Geomatic Engineering, University of Mines and Technology, Tarkwa, Box 237, Ghana
5   [2]Centre for Water Resources Research, University of KwaZulu-Natal, PBAG x01, 3209, Scottsville, Pietermaritzburg, South Africa
[3]South African Environmental Observation Network, Grasslands-Wetlands-Forests Node, PO Box 13053, Cascades, 3202, Pietermaritzburg, South Africa

10   *Correspondence to*: Michael S. Aduah (msaduah@umat.edu.gh; soakodan@yahoo.com)

**Abstract.** This study analysed the separate and the combined impacts of climate and land use changes on hydrology on the Bonsa catchment in Ghana, West Africa, using the ACRU hydrological model. The study used five RCP8.5 climate change scenarios (wet, 25th percentile, 75th percentile, dry and a multi-model median of nine GCMs) from the CMIP5 AR5 models for near (2020 – 2039) and far (2060 – 2079) future time slices. Change factors were used to downscale the GCM scenarios to 15   the local scale, using observed climate data for the control period of 1990 to 2009. The land use of 1991 and 2011 were used as the baseline and current land use as well as three future land use scenarios (BAU, EG, EGR) for two time slices (2030 and 2070) were used. The study showed that under all separate climate change scenarios, overall flows reduced, but under combined climate and land use changes, streamflows increased. Under the combined scenarios, streamflow responses due to the different future land use scenarios were not substantially different. Also, land use is the dominant controlling factor in 20   streamflow changes in the Bonsa catchment under a dry climate change, but under a wet climate change, climate controls streamflow changes. The spatial variability of catchment streamflow changes under combined land use and climate changes were greater than the spatial variability of streamflow changes under climate change. The range of plausible future streamflows changes derived in this study provides natural resources and environmental managers of the Bonsa catchment, the first ever and the most current information to develop suitable adaptation and mitigation strategies, to prepare adequately for climate 25   and land use changes.

**Key words**: Bonsa catchment, change factors, climate changes, Ghana, land use changes, spatial variability



# 1 Introduction

Climate change has become a topical issue because of its far reaching impacts on economies, life, property and the environment. Climate change intensifies the hydrological cycle, with consequences such as increases in frequency and severity of droughts and floods. The impacts of climate change on hydrology influences agriculture, water supply, environmental sustainability and protection from floods and infrastructure. Therefore determination of the impacts of the climate change on hydrology at the local scale is urgently needed in order to promote sustainable development and the protection of life, property and the environment. Land use also plays an important role in the environment, as it partitions rainfall into the components of the hydrological cycle (Costa et al., 2003;D'Orgeval and Polcher, 2008;Warburton et al., 2010), such as evaporation, runoff and groundwater. Land use changes thus change the balance between the components of the hydrological cycle, which can lead to several challenges in water resources and environmental management. It has also been suggested (Buytaert et al., 2010) that these challenges are more variable at the local and regional scale than at the global scale. Furthermore, since climatic changes can influence land use changes (Warburton et al., 2012) and land use changes through vegetation dynamics can also feedback to impact regional climate (Wang and Eltahir, 2000;Xue, 1997), comprehensive understanding of land use change and climate change impacts on hydrology can only be gained from determination of both their combined and separate impacts. The combined effects of climate change and land use changes on hydrology and the environment can lead to severe water resources and environmental problems at the local scale.

In West Africa, during the past four decades, there has been extensive land use changes and intensification of climate change. For example, between 1990 and 2010, deforestation has resulted in the removal of about 32 million ha of forest (FAO, 2010) and during the twentieth century mean annual temperatures increased by $2^0$C, while mean annual precipitation decreased by approximately 20% (Hulme et al., 2001). According to the IPCC's assessment reports (IPCC, 2007, 2013), temperatures will continue to rise in the 21$^{st}$ century, but there is no clear trend in precipitation for the region. It has been argued that the changes in the past four decades have largely been the result of the large scale drought in West Africa, between the 1970s and the early 1980s (Leblanc et al., 2008;L'Hote et al., 2002), agricultural expansion, as well as population increase and urbanization (Barbier, 2000;Jalloh et al., 2013).

Despite the substantial changes in West Africa and the numerous studies that have been conducted (Ardoin-Bardin et al., 2009;Bossa et al., 2014;Bossa et al., 2012;Ruelland et al., 2012;Sood et al., 2013), the responses to climate change and land use changes are still not understood well, especially in the southern rainforest catchments, where the majority of the population live. Aduah (2016) concluded that land use changes have altered the hydrology and the ecology of the Bonsa catchment of Ghana (a representative rainforest catchment of West Africa), with both increases in peak and low flows. The future scenarios of land use change in the catchment point to higher increases in both peak and low flows and a higher potential for ecological alterations. However, since the study did not consider effects of climate, it is not known how climate change and the combined



impacts of land use changes and climate changes will affect the hydrological cycle components in the near and far future. Since land use and climate changes affect each other (Dale, 1997;D'Orgeval and Polcher, 2008) and their joint impacts are sometimes non-linear (Li et al., 2009), it is necessary to determine both the joint and separate impacts on the hydrology of the catchment to improve knowledge and understanding of global change impacts in the region.

The aims of this study were to determine the impacts of climate changes, as well as the combined impacts of climate change and land use changes on the hydrology of the Bonsa catchment, which was selected to represent lowland rainforest catchments of West Africa. The study builds on a companion study in Aduah (2016) and it uses the ACRU hydrological model, which has previously been calibrated (Aduah et al., 2017) for the study area.

## 2 Methodology and Data Acquisition

### 2.1 Geographical Overview

The Bonsa catchment in Ghana, West Africa (Figure 1) is located between longitudes 1° 41′ and 2° 13′ West and latitudes 5° 4′and 5° 43′North. The catchment (1482 km$^2$) has a low relief, with elevation ranging between 30 and 340 m above mean sea level. The rainfall regime is bimodal, with the major season peaking in July and the minor season peaking in October. The annual rainfall ranges between 1500 mm and 2150 mm and the annual average minimum and maximum temperatures are 23ºC and 31ºC, respectively. The catchment land cover consists mostly of evergreen and secondary forests and shrubs/farms (Aduah et al., 2015). The geology is characterized by Tarkwaian and Birimian rock systems (Akabzaa et al., 2009) and the soil is composed mostly of Ferric Acrisols. Major economic activities in the catchment include open-pit gold mining, rubber and small-scale cocoa and food crop cultivation.

### 2.2 The ACRU Hydrological Model

The ACRU Model (Schulze, 1995) is a daily time step physical-conceptual hydrological model developed by the University of KwaZulu-Natal, South Africa to simulate catchment hydrological responses. It is a multi-purpose and multi-layer model (Figure 2) that can be used for catchment water resources assessment, assessment of land use change (Schulze, 2000;Schmidt et al., 2009;Warburton et al., 2012) and climate change impacts (Forbes et al., 2011;Graham et al., 2011;Kienzle et al., 2012). A detailed description of the model is available in Schulze (1995). The ACRU model was selected for this study because a satisfactory monthly NSE of 0.6 and R$^2$ of 0.7 were obtained after a physically meaningful calibration study undertaken for the Bonsa catchment (Aduah et al., 2017) and because the model has also been applied successfully in a variety of catchments (Forbes et al., 2011;Ghile, 2004;Warburton et al., 2010), which have a wide range of land uses and climates.



### 2.3 Model Data Acquisition and Preparation

A total of 103 subcatchments were created using digitized contour and river courses maps obtained from the Survey of Ghana
(SOG). The associated hydrological response units (HRUs) were created based on catchment land uses (Aduah et al., 2015).
In order to define the flow path of the rivers, a catchment configuration network, using subcatchments, HRUs and the river
courses, was created. The land use data, soil information and streamflow simulation control variables used in this study are the
same as those used in the companion papers in (Aduah et al., 2017). To determine the impacts of climate and land use change
on the hydrological responses of the Bonsa catchment various simulations were performed. The calibrated ACRU hydrological
model was run for the Bonsa catchment for the baseline scenario, using the baseline land use of 1991 and baseline observed
climate data from 1990 to 2009. The separate impacts of climate changes on streamflows were determined by running the
ACRU model for each selected climate change scenario and holding the land use as the baseline (1991) land use. Following
this, the combined impacts of climate and land use change were determined by varying both the climate change and the land
use scenarios.

**2.3.1 Land Use Scenarios**

In addition to the baseline (1991) and current (2011) land use, three future land use scenarios (Figure 3 and Table 1), namely
business-as-usual (BAU), economic growth (EG) and economic growth with enhanced reforestation (EGR), for two time slices
(2030 and 2070) were used. The land use scenarios were obtained from Aduah (2016).

**2.3.2 Global Climate Model Scenario Selection and Downscaling**

Global climate models (GCMs) are the most important tools for studying climate change impacts (Fowler et al., 2007;Teng et
al., 2012). Substantial progress has been made in GCM modelling in the past several decades, including the use of finer
resolutions and realistic parameterisation of vegetation and many physical processes (Alo and Wang, 2010;IPCC, 2007, 2013).
Unfortunately there are still numerous sources of uncertainties in GCM climate data, especially at the regional scale, due to
the inability of the models to capture small scale processes accurately (Buytaert et al., 2010), coarse spatial resolution (Quintana
Segui et al., 2010;Teng et al., 2012), uncertain future emission scenarios (Kim et al., 2013), inconsistent rainfall estimation
(Moradkhani et al., 2010) and biases in extreme events (Teng et al., 2012). Thus GCM climate data is still unsuitable for
hydrological modelling at the local scale (Buytaert et al., 2010). For impact assessments, such as the current study, downscaling
to a finer resolution (Buytaert et al., 2010;Liu et al., 2008) using either dynamic or statistical downscaling (SD) (Kunstmann
et al., 2004;Maurer and Hidalgo, 2008) is  required.



Consequently, statistically downscaled future climate records of the 8.5 Representative Concentration Pathway (RCP) scenario from four GCMs and a multi-model median of all available nine CMIP5 AR5 models were obtained from the Climate System Analysis Group (CSAG, 2015) of the University of Cape Town, South Africa. The selection of GCMs was based on changes

in mean annual precipitation (MAP) (Table 2 and 3). Two GCMs representing the extremes (wettest:CNRM-CM5 and driest: MIROC-ESM) and two representing 75[th] (GFDL-ESM2M) and 25[th] (MIROC5) percentile changes in MAP as well as a multi-model median of available downscaled GCMs were used for the study. The selection of the wettest and driest GCMs was to account for hydrological extremes (Lutz et al., 2016) such as floods and droughts, while the selection of the 75[th] and the 25[th] percentiles of GCMs was to avoid possible outliers in the climate projections (Mendlik and Gobiet, 2016) to provide a more

robust impact analysis. The use of the multi-model median of the GCMs for the study was to account for average hydrological conditions, which are also vital for sustainable water supply as well as proper functioning of the environment. The methods of selection of the GCMs (Forbes et al., 2011;Kienzle et al., 2012;Lutz et al., 2016;Mango et al., 2011;Mendlik and Gobiet, 2016;Moradkhani et al., 2010) was to support analysis of the full range of impacts, without using all available GCMs.

The selected GCMs were already downscaled to a few synoptic climate stations in Africa by CSAG (2015) as part of the World Climate Research Programme (WCRP) (Kim et al., 2013) Coordinated Regional Climate Downscaling Experiment for Africa (CORDEX-Africa). The scenario datasets used for the study formed part of the Coupled-Model Inter-comparison Phase five (CMIP5) project, which were used for the Fifth Assessment Report (AR5) of the IPCC (IPCC, 2013). The RCP8.5 scenario was selected because it prescribes a continual global warming into the 21[st] century through increasing radiative forcing and

greenhouse gas emissions (Moss et al., 2010), which is consistent with historical climate trends in West Africa (Hulme et al., 2001). Based on recent analysis (Jalloh et al., 2013), this trend of environmental degradation and climate change may continue to 2100, increasing the greenhouse gas emissions from the region, making the RCP8.5 the most suitable climate scenario for the region.

Since the CSAG data had no downscaled climate stations within the Bonsa catchment, three nearby and surrounding synoptic climate stations (Figure 1) in Accra (Ghana), Adiake and Bondoukou (Ivory Coast), which had downscaled GCM climate records were selected and the monthly mean of rainfall, minimum and maximum temperatures were computed and used for further downscaling to the Bonsa catchment using the change factor method of statistical downscaling (Chen et al., 2011;Fowler et al., 2007). For the rainfall, the change factor was calculated by dividing the monthly mean of each selected

GCM, as well as the multi-model median, for the near (2020-2039) and far (2060-2079) future by the corresponding monthly values for the control period (1990-2009). The change factor at the monthly time scale was then applied to the daily observed rainfall records of Bonsa catchment for the control period (1990-2009) to obtain climate change scenarios (Figure 4). For the minimum and maximum temperatures, the near and far future values were subtracted from their corresponding control period values and the difference was applied to the observed daily records.




The change factor method of downscaling is the most widely used method for impact analysis (Chen et al., 2011), however, its limitation is that it assumes that the variability of the future climate will be same as the control observed climate (Forbes et al., 2011) and in applying the method relative climate changes are assumed to be more important than absolute changes. The change factor method is therefore limited as it does not account for possible changes in intensity, duration and frequencies of

rainfall events (Ruelland et al., 2012). Thus the results obtained in this study are only a contextualized scenario of the potential climate change impacts on long-term annual and seasonal hydrology. More suitable downscaling methods such as spatially disaggregated and random cascades (Groppelli et al., 2011;Sharma et al., 2007) and Regional Climate Modelling (Kim et al., 2013) could have been used, but for lack of long-term predictor data and computational resources.

## 3 Results

### 3.1 Climate Change Impacts

Climate changes in the Bonsa catchment under the selected scenarios generally lead to increased temperatures. For rainfall, however, the changes were consistent but varied with seasons. Reductions in rainfalls were evident for the first half of the year, with increases evident for the second half of the year (Figure 4). Projections in rainfall for the GCM scenarios with the 25$^{th}$ and 75$^{th}$ percentile change in MAP were inconsistent, compared with the other scenarios as the differences between near

future monthly rainfalls for the 25$^{th}$ and the 75$^{th}$ percentile scenarios were not much. The sections below describe the temporal dynamics and the spatial variability of the impacts of the selected climate change scenarios on streamflows of the Bonsa catchment.

### 3.1.1 Temporal Dynamics of Climate Change Impacts on Hydrology

Annual streamflows, monthly median flows and high flows generally reduced for the near future, but increased for the far future climate change scenarios (Table 4, Figures 5, 6 and 7). The major peak season flows reduced in all the time slices (Table 4) and resulted in reduced low flows (Figure 6). The wet scenarios generated the highest streamflows, while the dry scenarios produced the lowest. The results in Table 4b for the near future time slice are inconsistent, compared with those of Table 4a, as streamflow changes for the 75$^{th}$ percentile GCM were less than those of the 25$^{th}$ percentile GCM. However, for accumulated

mean monthly streamflows, the trends for all the scenarios were similar (Figure 5).

The results generally appear to indicate longer dry seasons particularly for the far future and shorter wet seasons, despite the higher magnitudes of the peak season flows (Figure 5 and Figure 6). The lengthening of the dry season, shortening of the rainy season and increase in flows during the minor peak season could reduce agriculture productivity in the Bonsa catchment as the

growing period may reduce and the risk of flooding during the minor peak season will be higher than it currently is.  Hence, impacts of climate change could be a food security challenge in the catchment.



### 3.1.2 Spatial Patterns of Climate Change Impacts on Hydrology

An example of how streamflow responses to climate changes are distributed within the Bonsa catchment under the near future
(2020 - 2039) wet and dry GCM scenarios is shown in Figure 8. The results for the 75th and 25th percentiles GCM scenarios
are not shown as the pattern of streamflow changes were similar in all the climate scenarios, with the exception of the near
future streamflow projections for the 25th percentile GCM scenario (Table 4b). For the wet scenario, the streamflow changes
ranged from no change to a 5 - 25% increase, while for the dry GCM scenario there was 5 - 27% streamflow reduction. It is
evident that subcatchments in the main stem of the river experienced similar change (+5 - 15%) in streamflows under the wet
scenario, but under the dry scenario these subcatchments showed a 15-27% reduction in streamflow (Figure 8). These
subcatchments constituted more than 50% of the catchment, hence they influenced the outlet streamflows in the respective
scenarios.

### 3.2 Combined Impacts of Climate and Land use Changes

**3.2.1 Temporal Dynamics of Combined Future Climate/Land Use Change Impacts**

The annual, seasonal as well as high, median and low streamflows of the Bonsa catchment increased under the combined future
land use and climate change scenarios (Table 5, Figures 9 and 10). The differences in streamflows associated with the future
land use scenarios were small, with larger differences between the climate change scenarios (Figure 11). The streamflows
associated with the BAU combined land use and climate change scenarios were slightly higher than those associated with the
EG and EGR scenarios. The combined scenarios associated with the wet climate change scenario generated the highest
streamflows, while those associated with the dry climate change scenarios generated the lowest streamflows, regardless of the
time slice and land use involved (Table 5, Figures 9, 10 and 11). Minor reductions (up to 8%) in streamflows occurred in the
major peak season (Table 5) with higher increases in flows occurring in the minor peak season.

Under wet scenarios, the combined impacts had higher increases in annual streamflows than those under the separate land use
change impacts, but under a dry climate scenario, the combined impacts were lower than those under separate land use changes,
regardless of the time slice (Table 5). As a result of inconsistent rainfall projections, the near future streamflow changes (values
in parentheses) for the 25th percentile GCM were lower than those of the 75th percentile GCM. The results also indicate that
the combined impacts of land use and climate change in the Bonsa catchment are additive of the separate impacts. For example,
the change in annual streamflows for the BAU land use scenario of 32.1% and the corresponding impacts under the multi-
model climate change scenario (2020-2039) of -4.8%, sum closely to the annual change in combined impacts under the multi-
model 2020-2039 and BAU land use of 27.7% (Table 5).



### 3.2.2 Spatial Patterns of Combined Climate/Land Use Change Impacts

The spatial response to combined land use and climate change for the near future time slice under a wet and a dry climate
scenario is shown in Figure 12. The use of the wet and dry scenarios is to demonstrate how the availability of water affect the
spatial distribution of combined land use and climate change impacts in Bonsa catchment as the wet and dry scenarios can
represent spatial patterns in streamflow changes similar to the rest of the combined scenarios. Under the BAU, EG and the
EGR land uses and wet climate scenario, subcatchments along the main stem of the Bonsa River experienced streamflow

changes between 5 - >75% for the northern and 40-55% for the southern areas. The increase in streamflows along the main
stem of the river is because the land use changed from secondary forests in 1991 to shrubs/farms in the future land use scenarios
(Figure 3). For subcatchment 88 and its surrounding areas in the west central part of the catchment, streamflow increases
ranged from 45-65% under the BAU, EG and the EGR scenarios with the wet climate. The large increase in streamflows in
this west central part of the catchment was due to the expansion of the settlements (Tarkwa), mining areas (east and west of

Tarkwa) and increased shrubs/farms in the future scenarios, compared to the baseline 1991 land use (Figure 3).

For those subcatchments in the southwest, the middle and the northern parts of the catchment, which remain as evergreen
forest in the future land use scenarios (Figure 3), there was 5-25% increase in simulated streamflows under the scenarios of
future land use and a wet climate. There was also a 5-15% increase in streamflows (Figure 12) in the eastern tip, north western

tip and the northern tip of the catchment in subcatchments which were mostly covered by evergreen forests in both the historical
land use and the future land uses (Figure 3). In all the scenarios involving the wet climate, subcatchment 54 and its surrounding
areas had the highest overall increases in streamflows. The outlet streamflows with respect to the BAU and wet climate scenario
increased between 45-55%, while those with respect to the EG and the EGR increased by between 35-45% only (Figure 11).
The difference between the land use scenarios under wet climate change is that the BAU show higher magnitudes of increases

in streamflows for subcatchments 88, 54 and their surrounding areas, compared to the EG and the EGR land uses.

Figure 12 also show that the various land use scenarios under the dry climate change exhibited similar patterns as the
streamflow changes under the wet climate change, except that there were streamflow reductions (between 5-26%) in
subcatchments which either experienced no changes or reductions, under the wet climate and land use scenarios. The

streamflow increases along the main stem of the river under the dry climate also appear to dampen towards the outlet of Bonsa
catchment. It can therefore be deduced from Figure 12 that the variability of streamflow changes within the subcatchments
increased as land use and climate changes occurred simultaneously, but the variability of the streamflow changes at the outlet
of the catchment does not experience significant changes under combined land use and climate changes.



## 4 Discussion

### 4.1 Combined Climate and Land Use Change Impacts on Hydrology

The study analysed the separate impacts of climate change, as well as the combined impacts of climate and land use change on the hydrology of the Bonsa catchment, Ghana, West Africa, which to the authors' knowledge is the first time such a study has been conducted in the catchment. The study revealed that the effects of climate change for both the near and far future suggests an overall drying trend in the catchment, regardless of the scenario. The results also indicate a lengthening of the dry season in the Bonsa catchment in the far future, which can affect water availability for domestic, industrial and environmental flows.

On the other hand, streamflow increases due to combined land use and climate changes were higher than the forested 1991 baseline flows. During high flow periods, combined impacts analysis shows that high flow magnitudes generally increased beyond both baseline and current (2011) levels as did the low flows, regardless of the land use and climate scenario. The combined scenarios involving the wet, $25^{th}$ percentile, $75^{th}$ percentile and multi-model median climate scenarios showed slightly different patterns of changes in the length of the dry and major peak seasons. In the near future the duration and the onset of the dry season and the major peak season did not change for the wet, the $25^{th}$ percentile, the $75^{th}$ percentile and the multi-model median climate scenarios, but in the far future the dry season length increased, while the major peak season length reduced for the $25^{th}$ percentile, the $75^{th}$ percentile and the multi-model median climate scenarios (Figure 9). For the combined scenarios involving the dry climate scenario, there was a small shift in both the dry and major peak season lengths, as well as their onsets.

The streamflow responses for the combined scenarios involving the different land use scenarios (BAU, EG and EGR) were not very different, although the scenarios involving the BAU had slightly higher streamflows than the rest. This is mainly because although the secondary forest areas in the potential future land uses were different, the proportion of the shrubs/farms, which is the dominant land use in terms of both area and runoff generation, were not substantially different between the scenarios. Given the significant land use changes that already occurred between 1991 and 2011, the land use changes in the future in comparison will not significantly alter outlet streamflows assuming land cover changes as predicted by the model.

The study showed that the combined impact of land use and climate changes are additive of the separate impacts, unlike studies such as Li et al. (2009) where the joint impacts were non-linear. The study further shows that when there is substantial increase in rainfall (under the wet climate scenario), climate largely controls streamflow changes, but when there is less rain (under dry





climate scenario), land use controls streamflow changes. This is because under the wet climate and land use change, the impacts were higher than those under separate land use change, but under a dry climate and land use change, the impacts were higher than those under separate land use change (Table 6). Furthermore, the study revealed that the variability of climate change impacts at the catchment and subcatchment scales were also almost the same (Figure 8 and Table 4), as the streamflow changes at the two scales did not differ much. The study also reveals that variability of streamflow changes was greater at the

subcatchment than at the catchment scale under simultaneous land use and climate change. Hence, it is relevant to not only focus on the outlet streamflows when considering Global change impacts, and basing catchment management plans on outlet streamflow changes alone can result in less effective adaptation measures.

These results contradict those of previous studies in West Africa, where Bossa et al. (2014) concluded that streamflow

responses to combined impacts of climate change and land use changes were higher than climate change impacts, but less than separate land use change impacts. Bossa et al. (2012), also concluded that climate was the dominant factor in streamflow changes in a savannah dominated catchment in Benin Republic. Hence, the effects of Global changes revealed in this study (rainforest catchment) and others (mainly savannah catchments) in West Africa, portray within-region differences of hydrological impacts under changing conditions and provides a platform for further studies. The results of this study also

disproves the hypothesis that joint impacts of land use and climate changes are non-linear (Li et al., 2009).

### 4.2 Uncertainties in Assessing Land Use and Climate Change Impacts

Semi-distributed hydrological models are the most widely used tools for studying the impacts of climate change and land use change on hydrology, separately and jointly in regions with diverse land uses and climates (Breuer et al., 2009;Forbes et al.,

2011;Legesse et al., 2010;Warburton et al., 2012). This is because semi-distributed models simulate the hydrological cycle physically, spatially and temporally, to generate information for effective land use planning and water management decisions. The use of the physical-conceptual and semi-distributed ACRU model in this study hinged on the fact that it is sensitive to both land use and climate changes (Schulze, 1995), as well as being able to generate hydrological responses both at the subcatchment and catchment scales.


Apart from the uncertainties with hydrological models, the uncertainties with climate change and land use input data also affects the impact assessments. Previous studies (Buytaert et al., 2010;Thompson et al., 2013) show that the uncertainties associated with GCM climate change scenarios are greater than those related to hydrological models. In West Africa, the rainfall estimation by climate models is particularly problematic, as some models predict increases in the coastal areas and

decreases in the semi-arid, others predict the opposite (Ardoin-Bardin et al., 2009). Since rainfall is the major driver of the hydrology of humid regions and is the most sensitive parameter in the ACRU model, the future rainfall estimates, as well as the use of the change factor downscaling method, constitute the most significant source of uncertainty in the simulated



streamflows in this study. However, as the study used five GCM climate scenarios (wet, 25th percentile, 75th percentile, dry and multi-model median) to quantify the uncertainties with climate changes, a contextualised scenario of alternative futures, upon which different catchment planning scenarios can be evaluated, have been provided. Another source of uncertainty with this study is the modelled future land uses. Since the land use modelling was based on historical land use data, the far future (2070) land use scenarios, are more uncertain, compared to the near (2030) future land uses. Although the use of modelled future land use introduces uncertainties into the simulated streamflows, land use modelling is the most realistic method to provide plausible future land use scenarios, as it relies on statistically significant socio-economic and biophysical driving factors. Land use modelling also ensures that gradual changes, including simultaneous regeneration and removal of land use types at different locations within a catchment, are accounted for. This ensures that realistic change processes, previously witnessed in a catchment are applied. This study used three land use scenarios (BAU, EG and EGR), which provide alternative development pathways for the catchment, to quantify the uncertainties with the land use.

Furthermore, uncertainties have also been introduced due to the use of a coarse resolution soil map (scale: 1: 250 000) and the lack of locally measured soil hydrological properties, as well as lack of vegetation parameters for the Bonsa catchment. These uncertainties have been minimized in this study since the sensitivity analysis and the hydrological modelling in the companion paper (Aduah et al., 2017) resulted in satisfactory calibration, making it possible to provide a first estimate of Global change impacts on hydrology in the Bonsa catchment.

## 5 Conclusions

This study has shown that in the Bonsa catchment under climate change, streamflows generally reduced, compared to the forested (1991) baseline conditions. Under combined land use and climate change scenarios, streamflow increases relative to the forested (1991) baseline, were higher, compared to the climate change scenarios, irrespective of the time slice. The impacts of simultaneous land use and climate changes on streamflows were higher than those under separate land use changes when there was considerable increase in rainfall, but lower when there was less rainfall. This means climate will control streamflow changes when there is substantial increase in rainfall, but land use will control the streamflows changes if rainfall reduces. The streamflow responses in the combined scenarios with respect to the different future land use (LU) scenarios were however, not substantially different, for a particular LU scenario to be selected over the other. In terms of spatial variability of Global change impacts, the streamflow changes had similar patterns especially along the main stem of the river. When land use and climate changes occur simultaneously, the streamflow increase were higher and the spatial variability of streamflow changes was higher than when only climate changes are considered.




The use of both multiple future land uses and climate change scenarios in the impact assessments has provided a range of first stage estimation of potential Global change impacts on the hydrology, which policy makers can use for land use and

environmental planning, as the various scenarios can be evaluated against a set of planning objectives for the catchment.

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





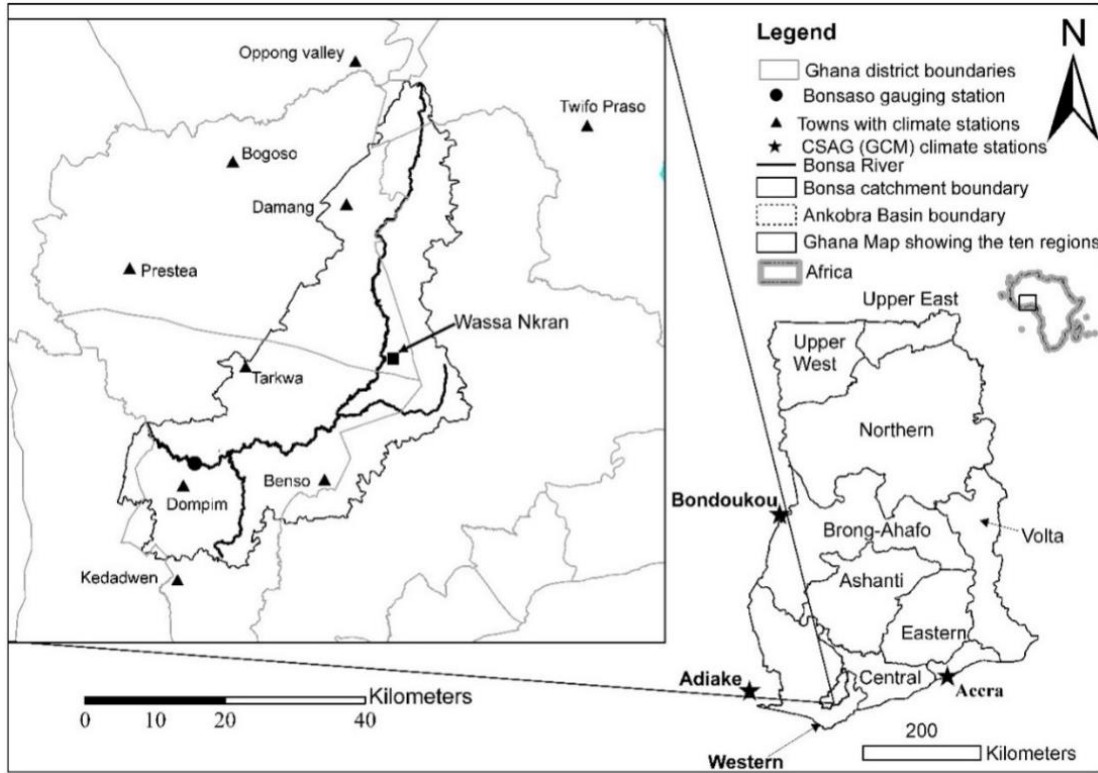

**Figure 1: Map of study area showing the Bonsa catchment in the Ankobra basin, Ghana and the GCM (CSAG) as well as observed climate stations.**





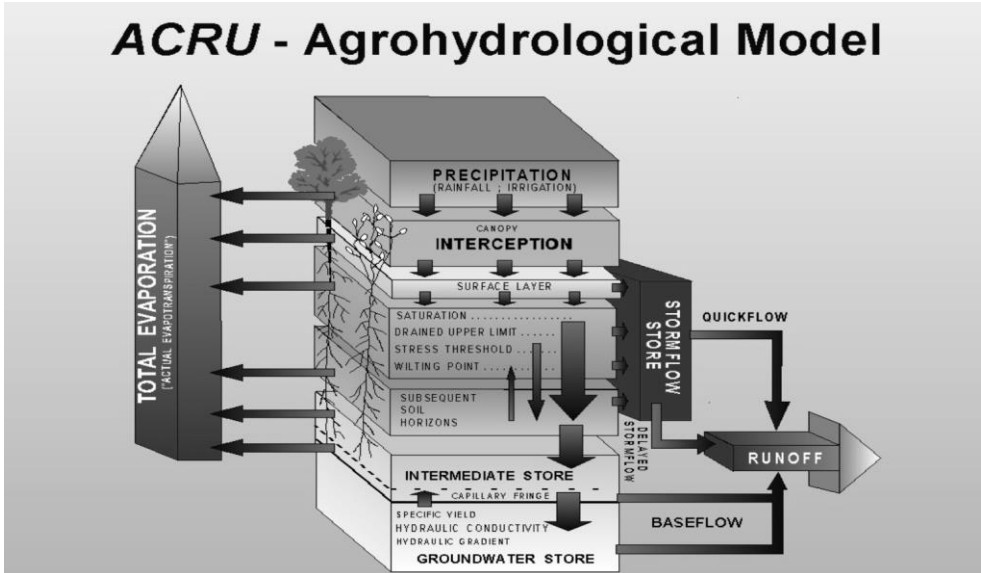

**Figure 2: Representation of hydrological processes in the ACRU model (Schulze, 1995).**

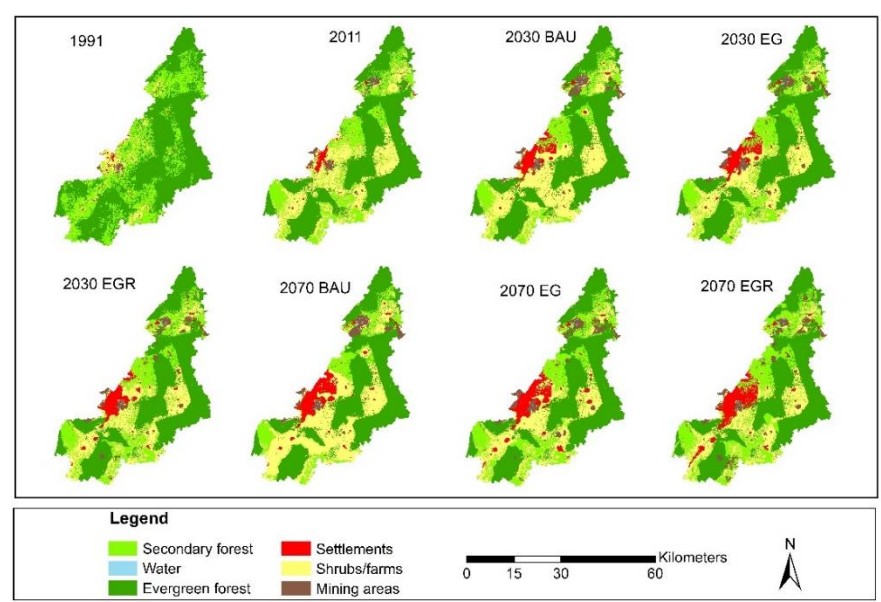

**Figure 3: Land use maps for baseline land use (1991), current (2011) land use and future land use scenarios in the Bonsa catchment, generated from mapping and land use simulation (Aduah, 2016).**


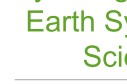
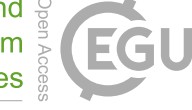



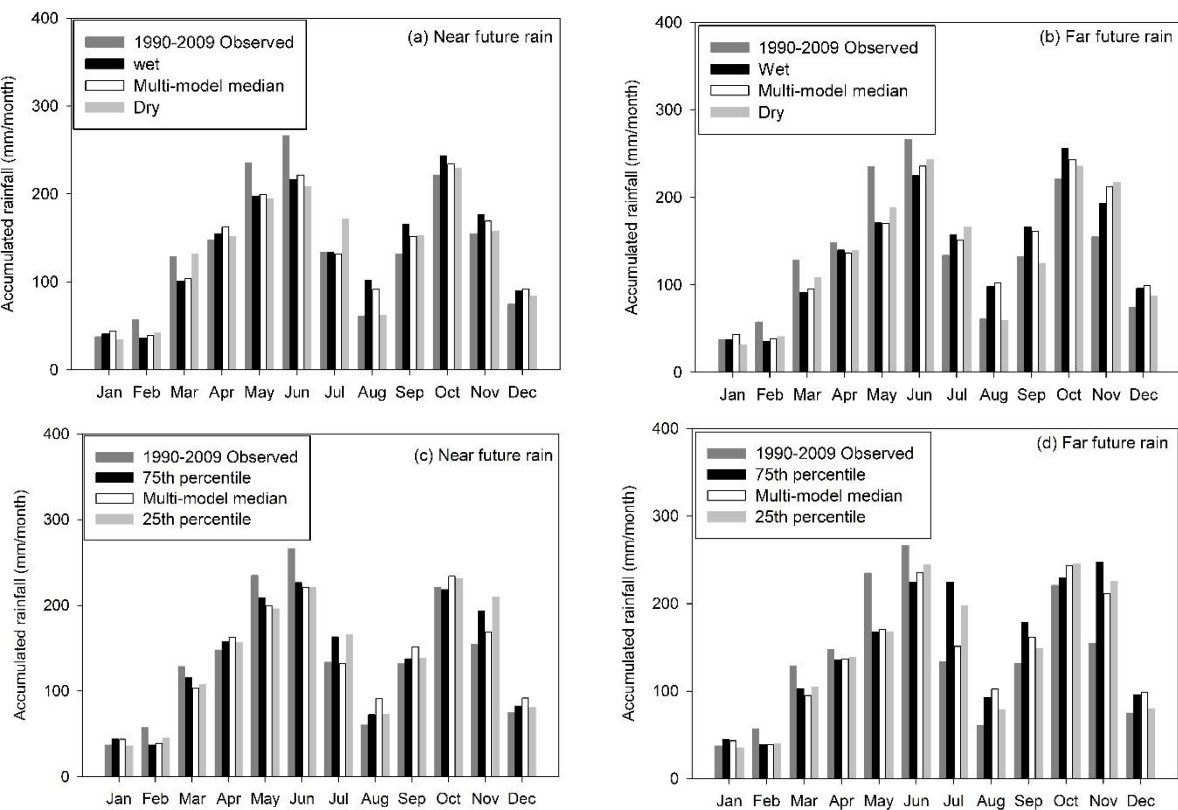

**Figure 4: Downscaled and observed monthly mean rainfall for Tarkwa climate station in Bonsa catchment.**



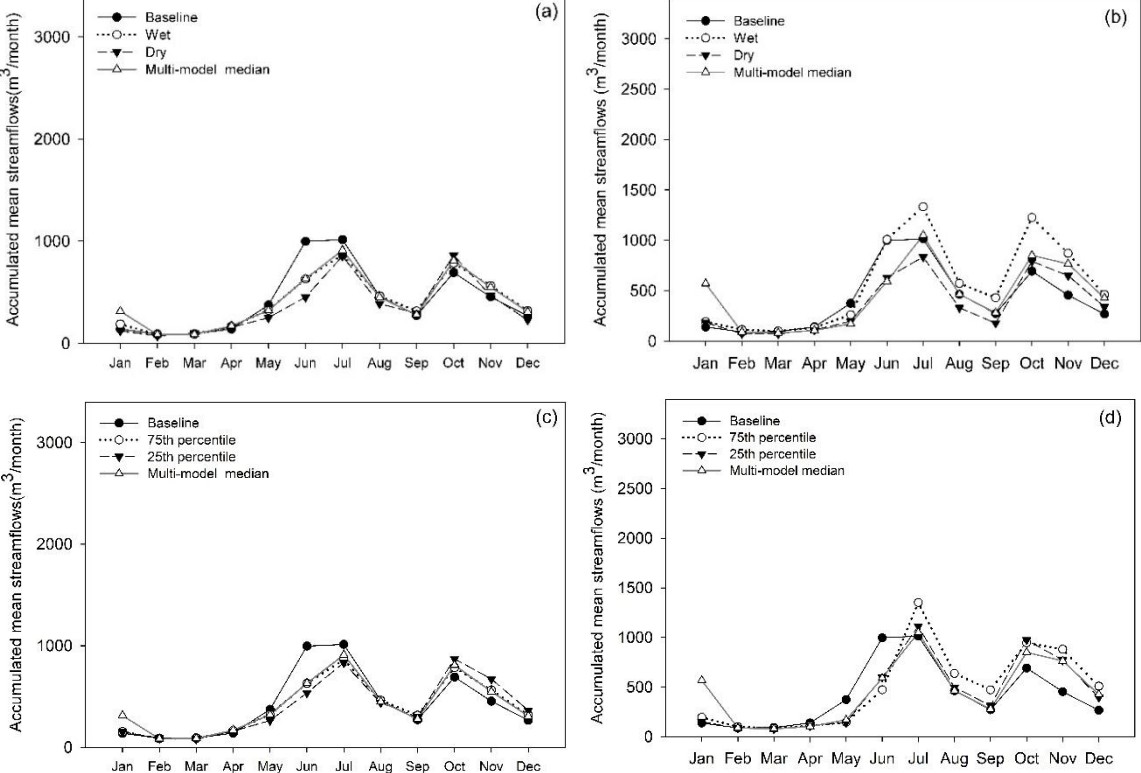

**Figure 5: Accumulated mean monthly streamflows for the selected climate change scenarios for the near future (2020 – 2039) (a, c)**
**and far future (2060 – 2079) (b, d), compared with the baseline streamflows.**



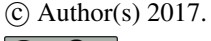

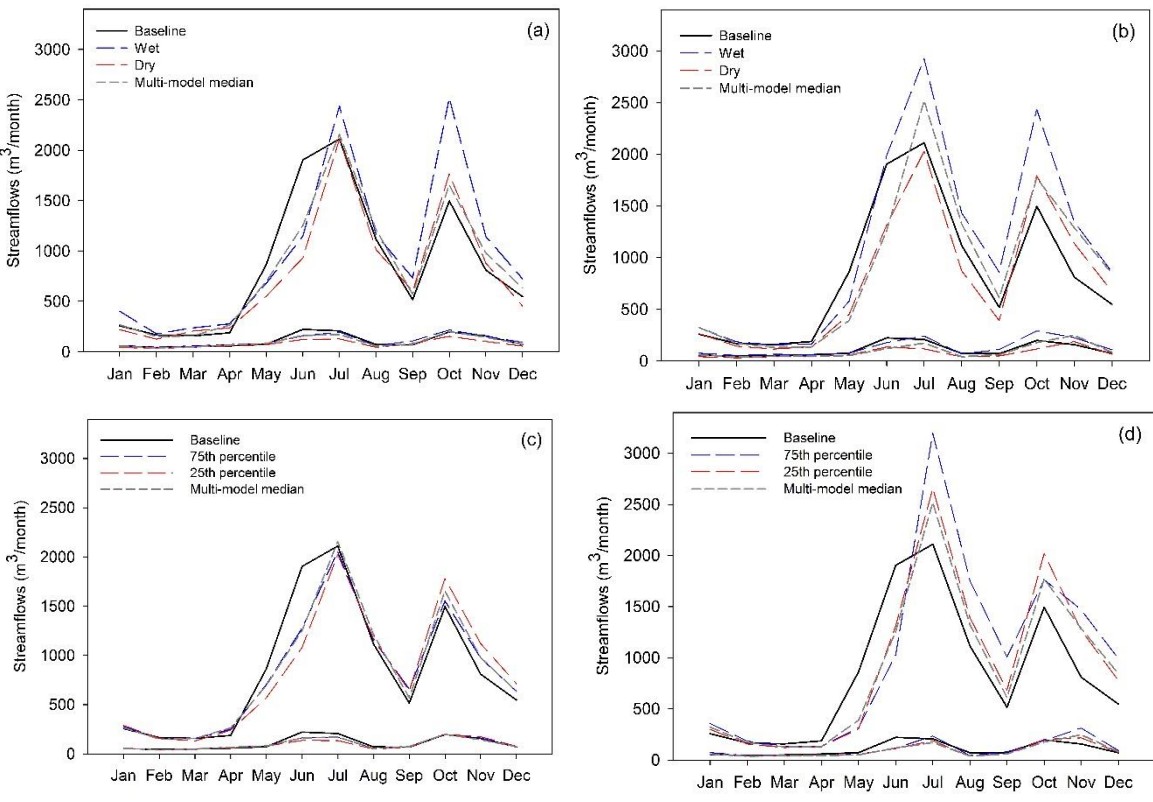

**Figure 6: Monthly 90th (1 in 10 year high) and 10th percentile (1 in 10 year low) streamflows for the selected climate scenarios for the near future (2020 – 2039) (a, c) and (b, d) far future (2060 – 2079), compared with the baseline streamflows.**

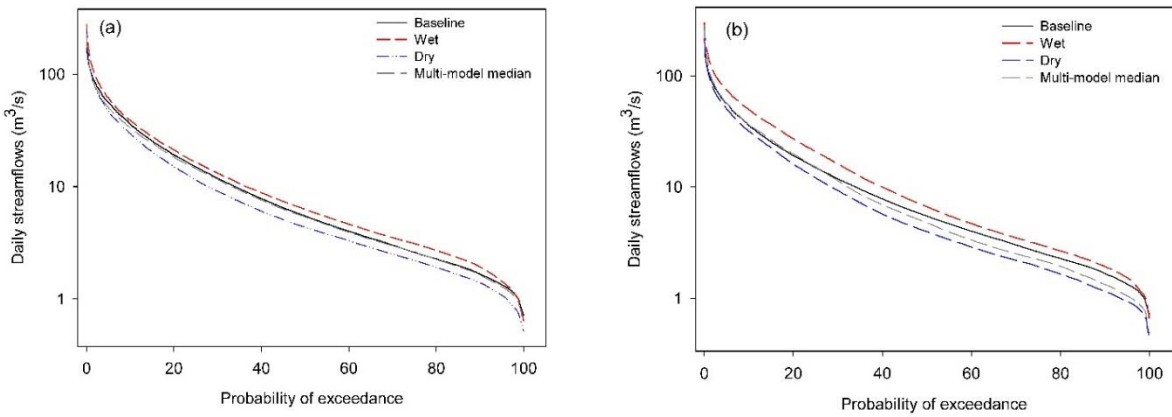

**Figure 7: Flow duration curves for the wet, dry and multi-model median climate change scenarios for two time slices (a) 2020 - 2039 and (b) 2060 – 2079, as well as the baseline FDCs.**





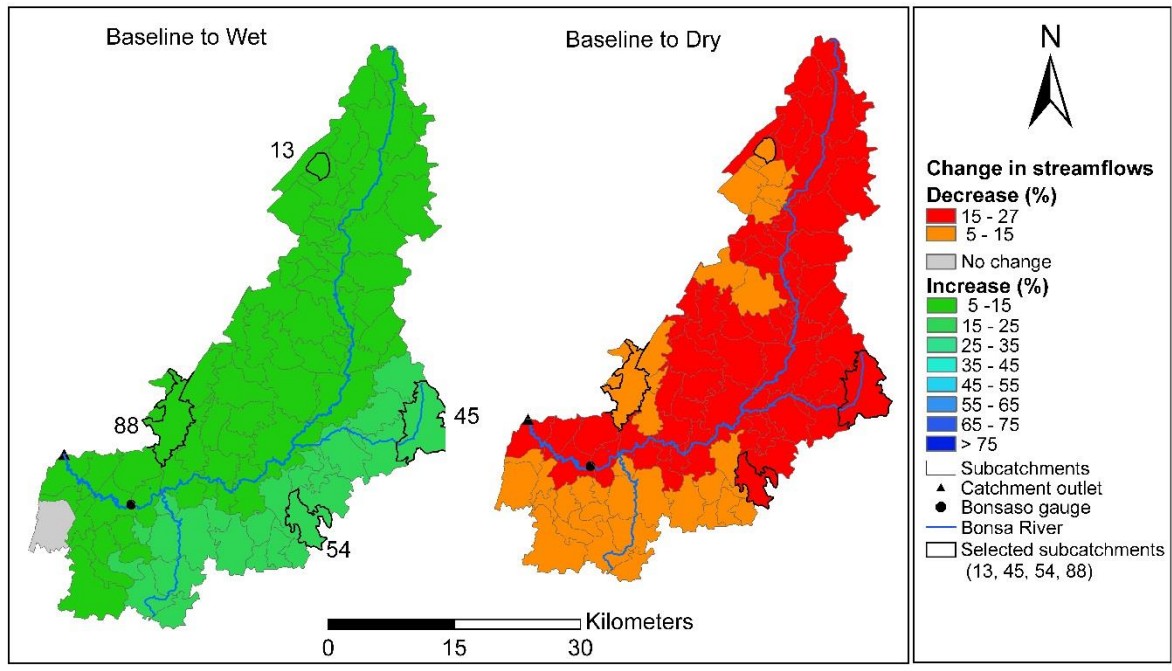

**Figure 8: Changes in mean annual streamflows relative to baseline (1991 land use and 1990-2009 climate) for 2020 - 2039 wet and dry climate change scenarios. Numbers in map indicate subcatchments.**





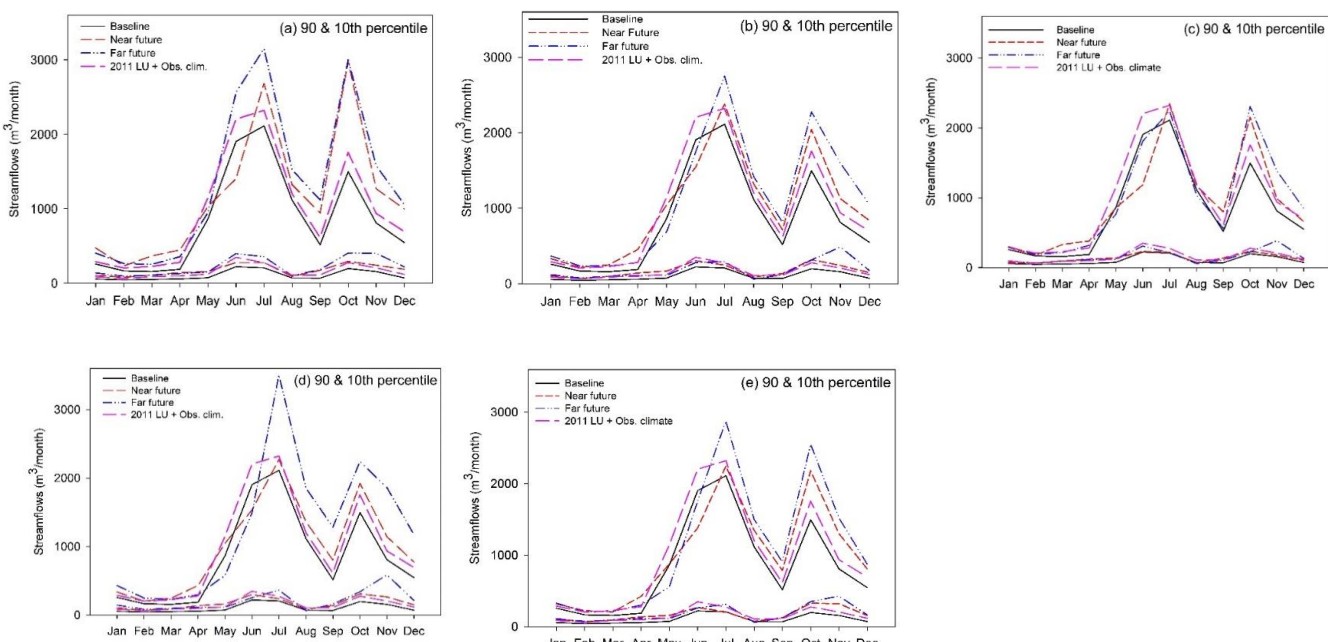

**Figure 9: Monthly 90th (1 in 10 year high) and 10th (1 in 10 year low) percentile streamflows for combined (a) BAU+ wet, (b) BAU+ multi-model median, ( c) BAU+ dry climate change scenarios, (d) BAU + 75th percentile GCM scenario and (e) BAU+25th percentile GCM scenario, as well as baseline (1991) and 2011 Land use (LU) +Observed climate (1990-2009).**





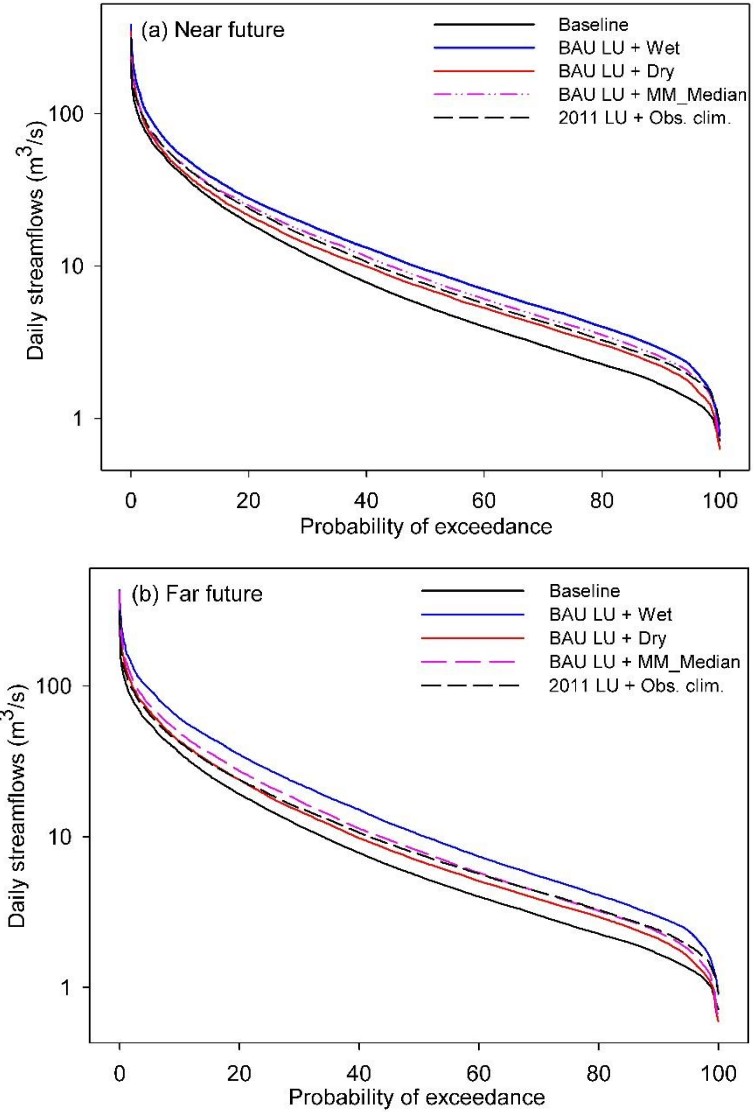

**Figure 10: FDCs for combined near future (a) 2020 – 2039 and far future (b) 2060 – 2079 GCM climate scenarios and the BAU land use scenario, compared to those of the baseline and current (2011) FDCs.**




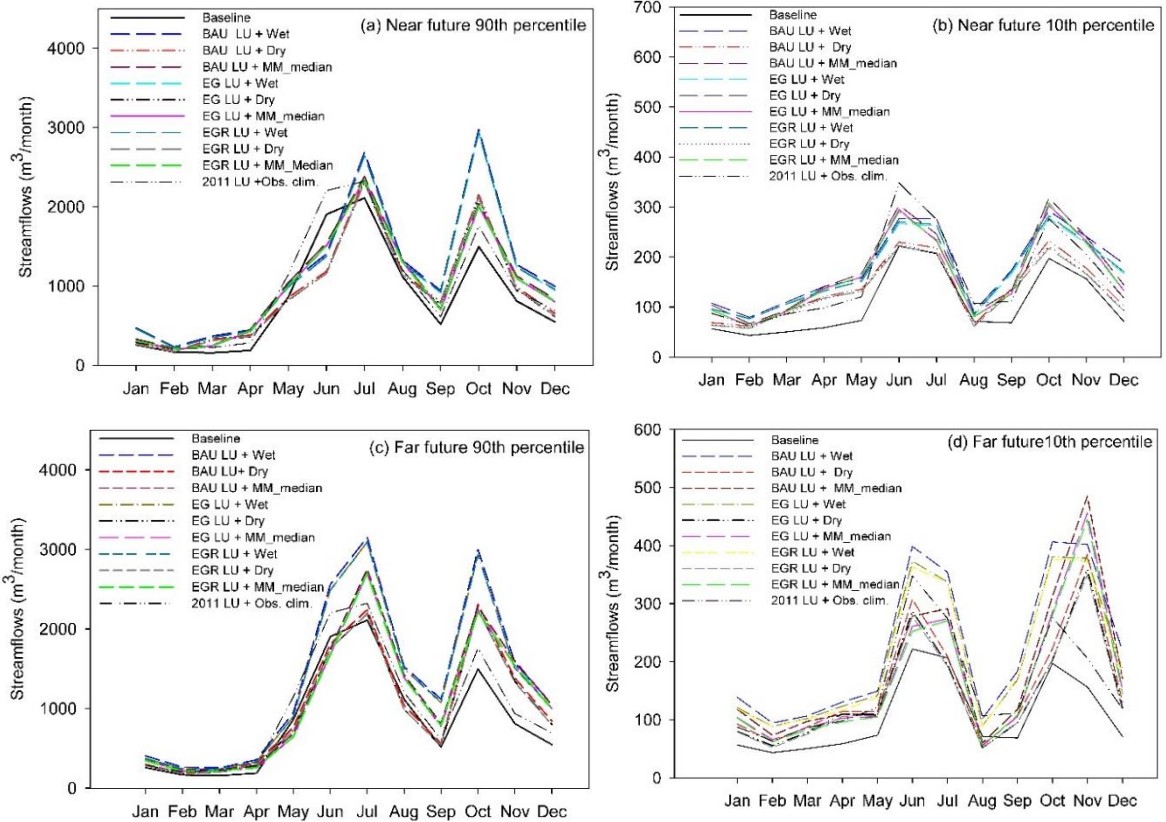

**Figure 11: Streamflows for combined impacts of land uses (BAU, EG, EGR) and wet, multi-model median (MMM) and dry climate change scenarios for (a) 90th percentile near future, (b) 10th percentile near future, (c) 90th percentile far future and (d) 10th percentile far future climate change scenarios. Figures also show the 90th and 10th percentile streamflows for the baseline and 2011 land use (LU) + Observed climate (1990-2009).**





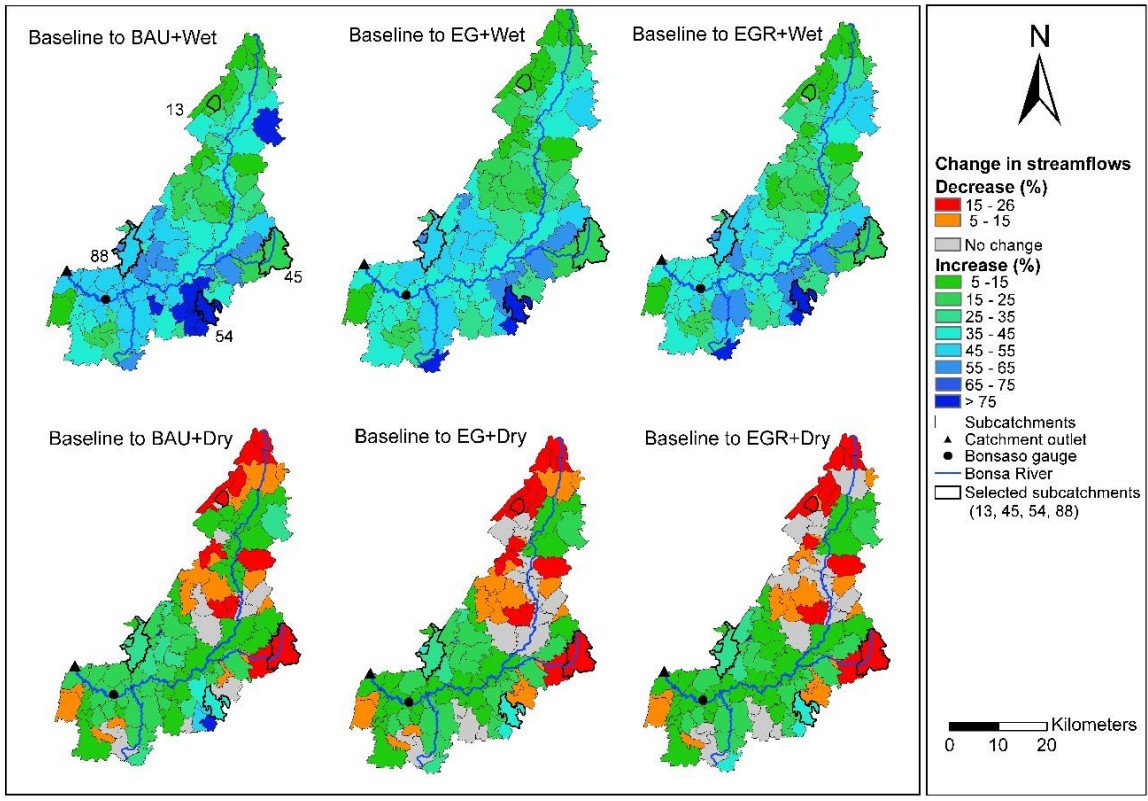

**Figure 12:** **Changes in mean annual streamflows relative to baseline (1990 - 2009) for combined land use change (2030) and 2020 - 2039 wet and dry climate change scenarios. Numbers in map indicate selected subcatchments.**





**Table 1: Proportions of land uses for the Bonsa catchment for the baseline (1991), current (2011) and future scenarios (2030 and 2070).**

| Scenario/ Time slice | Land cover (km$^2$) | | | | | | |
|---|---|---|---|---|---|---|---|
| | Secondary forest | Water | Evergreen forest | Settlements | Shrubs/ farms | Mining areas | Total |
| Baseline | 457.3(30.8) | 0.2(0.1) | 916.5(61.8) | 12.4(0.8) | 90.0(6.1) | 5.9(0.4) | 1482.3 |
| Current | 280.3(18.9) | 1.4(0.1) | 754.6(50.9) | 22.4(1.5) | 399.4(26.9) | 24.2(1.6) | 1482.3 |
| BAU:2030 | 215.5(14.5) | 1.4(0.1) | 728.1(49.1) | 63.7(4.3) | 425.3(28.7) | 48.3(3.3) | 1482.2 |
| 2070 | 171.0(11.5) | 1.4(0.1) | 683.6(46.1) | 91.3(6.2) | 482.1(32.5) | 52.8(3.6) | 1482.2 |
| EG: 2030 | 283.9(19.2) | 1.4(0.1) | 733.6(49.5) | 68.8(4.6) | 340.1(22.9) | 54.3(3.7) | 1482.1 |
| 2070 | 294.7(19.9) | 1.4(0.1) | 689.4(46.5) | 95.7(6.5) | 335.0(22.6) | 66.0(4.5) | 1482.2 |
| EGR:2030 | 303.8(20.5) | 1.4(0.1) | 716.0(48.3) | 66.6(4.5) | 340.2(22.9) | 54.3(3.7) | 1482.2 |
| 2070 | 350.1(23.6) | 1.4(0.1) | 667.7(45.0) | 100.6(6.8) | 295.4(19.9) | 67.2(4.5) | 1482.2 |

Numbers in brackets are percentages (%)

**Table 2: GCMs from IPCC CMIP5 AR5 used in this study.**

| No. | GCM | Modelling Centre | Scenarios (from CSAG) |
|---|---|---|---|
| 1 | MIROC-ESM* | Japan Agency for Marine-Earth Science and Technology, Atmosphere and Ocean Research Institute (The University of Tokyo), and National Institute for Environmental Studies | 4.5/8.5RCP |
| 2 | CNRM-CM5* | Centre National de Recherches Meteorologiques / Centre Europeen de Recherche et Formation Avancees en Calcul Scientifique | 4.5/8.5RCP |
| 3 | CanESM2 | Canadian Centre for Climate Modelling and Analysis | 4.5/8.5RCP |
| 4 | FGOALS-s2 | LASG, Institute of Atmospheric Physics, Chinese Academy of Sciences | 4.5/8.5RCP |
| 5 | BNU-ESM | College of Global Change and Earth System Science, Beijing Normal University | 4.5/8.5RCP |
| 6 | MIROC5* | Atmosphere and Ocean Research Institute (The University of Tokyo), National Institute for Environmental Studies, and Japan Agency for Marine-Earth Science and Technology | 4.5/8.5RCP |
| 7 | GFDL-ESM2G | Geophysical Fluid Dynamics Laboratory, USA | 4.5/8.5RCP |
| 8 | MIROC-ESM-CHEM | Japan Agency for Marine-Earth Science and Technology, Atmosphere and Ocean Research Institute (The University of Tokyo), and National Institute for Environmental Studies | 4.5/8.5RCP |
| 9 | GFDL-ESM2M* | Geophysical Fluid Dynamics Laboratory, USA | 4.5/8.5RCP |

* denotes GCMs used for study





**Table 3: Selected GCMs based on changes in projected MAP in 2060 - 2079 relative to the 1990 - 2009 control period for Adiake, Bondoukou and Accra in West Africa.**

| GCM | Change in MAP (%) | Variable lable |
|---|---|---|
| CNRM-CM5**** | 8.6 | Wet |
| FGOALS-s2 | 6.6 | |
| GFDL-ESM2M *** | 5.4 | 75th percentile |
| CanESM2 | 5.2 | |
| GFDL-ESM2G | 1.8 | |
| BNU-ESM | 1.7 | |
| MIROC5** | 1.1 | 25th percentile |
| MIROC-ESM-CHEM | 0.9 | |
| MIROC-ESM* | -3.8 | Dry |

\* denotes dry, \*\* 25th, \*\*\*75th and \*\*\*\* wet  GCMs

**Table 4: Change (%) in streamflows relative to the baseline (land use & climate) for all climate scenarios.**

**(a)**

| Time slice | Climate scenario/GCM | Change in streamflows (%) from baseline climate | | | |
|---|---|---|---|---|---|
| | | Annual | Dry season | Major peak | Minor peak |
| | Wet | 13.2 | 34.1 | -17.1 | 61.7 |
| 2020-2039 | Multi-model median | -4.8 | 6.9 | -19.4 | 15.5 |
| | Dry | -15.3 | -12.9 | -32.2 | 14.3 |
| | Wet | 33.9 | 46.0 | 8.2 | 77.9 |
| 2060-2079 | Multi-model median | 0.8 | 28.8 | -24.1 | 33.5 |
| | Dry | -13.6 | 2.6 | -30.0 | 13.8 |

**(b)**

| Time slice | Climate scenario/GCM | Change in streamflows (%) from baseline climate | | | |
|---|---|---|---|---|---|
| | | Annual | Dry season | Major peak | Minor peak |
| | 75th percentile | -4.8 | 9.5 | -21.4 | 17.3 |
| 2020-2039 | Multi-model median | -4.8 | 6.9 | -19.4 | 15.5 |
| | 25th percentile | -4.7 | 17.0 | -29.0 | 29.9 |
| | 75th percentile | 18.4 | 51.4 | -17.4 | 62.2 |
| 2060-2079 | Multi-model median | 0.8 | 28.8 | -24.1 | 33.5 |
| | 25th percentile | 4.8 | 20.9 | -22.2 | 45.4 |



**Table 5: Changes in streamflows relative to the baseline (1990-2009) for combined climate change/land use change scenarios as well as change in annual streamflows for separate land use and climate change impacts. Values in parentheses are for the 25th and 75th percentile GCMs. A: combined impacts, B: separate impacts, LUC: land use change, CC: climate change.**

| Land use | | Climate change | | A: Change in streamflows (%) | | | | B: Change in streamflows (%) | |
|---|---|---|---|---|---|---|---|---|---|
| Scenario | Year | Time slice | Scenario/GCM | Annual | Dry season | Major peak | Minor peak | LUC(Annual) | CC (Annual) |
| BAU | 2030 | 2020-2039 | Wet (75th) | 47.7 (27.6) | 88.7 (54.8) | 12.0 (7.1) | 102.4 (54.8) | | 13.2 (-4.8) |
| | | | Multi-model | 27.7 | 54 | 9.5 | 52.5 | 32.1 | -4.8 |
| | | | Dry (25th) | 16.0 (27.8) | 34.0 (61.1) | -5.4 (-1.5) | 50.7 (70.0) | | -15.3 (-4.7) |
| | 2070 | 2060-2070 | Wet (75th) | 76.5 (60.2) | 100.5 (108.6) | 44.8 (14.7) | 132.7 (119.1) | | 33.9 (18.4) |
| | | | Multi-model | 40.7 | 85.2 | 8.2 | 85.4 | 39 | 0.8 |
| | | | Dry (25th) | 24.2 (44.0) | 53.0 (71.0) | 2.4 (9.7) | 61.6 (98.0) | | -13.6 (4.8) |
| EG | 2030 | 2020-2039 | Wet (75th) | 43.2 (23.5) | 80.2 (47.5) | 8.7 (4.0) | 97.2 (50.1) | | |
| | | | Multi-model | 23.5 | 46.5 | 6.3 | 47.8 | 27.9 | |
| | | | Dry (25th) | 12.0 (23.6) | 26.9 (53.7) | -8.3 (-4.5) | 46.0 (65.0) | | |
| | 2070 | 2060-2070 | Wet (75th) | 69.2 (53.1) | 88.1 (95.1) | 39.6 (10.4) | 123.4 (109.6) | | |
| | | | Multi-model | 34 | 72.3 | 3.8 | 77 | 32.2 | |
| | | | Dry (25th) | 17.9 (37.5) | 41.5 (59.2) | -2.0 (5.4) | 54.0 (89.3) | | |
| EGR | 2030 | 2020-2039 | Wet (75th) | 43.4 (23.7) | 80.7 (48.0) | 8.8 (4.1) | 97.3 (50.2) | | |
| | | | Multi-model | 23.7 | 47.1 | 6.5 | 47.9 | 28.1 | |
| | | | Dry (25th) | 12.2 (23.8) | 27.3 (54.4) | -8.2 (-4.4) | 46.1 (65.1) | | |
| | 2070 | 2060-2070 | Wet (75th) | 68.1 (52.1) | 86.3 (93.2) | 38.9 (9.8) | 122.0 (108.2) | | |
| | | | Multi-model | 33 | 70.3 | 3.2 | 75.7 | 31.2 | |
| | | | Dry (25th) | 17.0 (36.5) | 39.6 (57.5) | -2.7 (4.8) | 52.8 (88.0) | | |