# Peer review of "Scenario-based impacts of land use and climate changes on the hydrology of a lowland rainforest catchment in Ghana, West Africa"

_Hydrology and Earth System Sciences, 2017_

## Referee Comment (RC1) · Anonymous Referee #1 · 1 Dec 2017

The manuscript "Scenario-based impacts of land use and climate changes on the hydrology of a lowland rainforest catchment in Ghana, West Africa" by Aduah et al. deals about the separate and combined analysis of impacts due to climate and luse use change using the ACRU model. For this, a calibrated model for the Bonsa catchment was used (Aduah et al., 2017, companion paper).

In general, impact studies for African catchments are of broad interest since adaptations plans regarding water management for the future will be necessary in the context of climate change. Hydrological or ecohydrological models can be a useful tool to support management decisions. However, the usefullness of model results strongly

[Figure]

Creative Commons BY license logo

depends on a reasonable application of the models and a thorough analysis of the model results. In this regard I have two main concerns that need to be addressed or clarified by the authors:

1. All presented results are based on modelled monthly streamflow. Model calibration was presented in a companion paper. It was reported that "validation based on the daily time step did not generate satisfactory performance as NSE of 0.14 and 0.31 were obtained during calibration and validation, respectively". As a consequence monthly time steps were used since performance was better.

In my opinion, it is not good modelling practice to leave out poor model performance on a daily time step and to present satisfying model performance on a monthly basis. There must be a reason for poor model performance and in this regard, the authors need to clarify, if this poor performance may have implications for all following applications and conclusions. How can the authors be sure that hydrological processes are adequately simulated? Obviously there are not well simulated. Otherwise, model performance for daily time steps were much better.

Additionally it is not good modelling practice to use only a small number of performance measures, especially if all selected measures are focused on peak and high flow. I respect the circumstance that the investigated catchment may be not intensively monitored and that data scarcity may be a problem. However, there are additional ways to make sure that the model behaves reasonable and of course realistically (e.g. using constraints, rules-of-thumb, multi-site calibration, model output such as discharge components or hydrological components over time).

2. Due to coarse temporal resolution, all derived conclusions do have a more general character such as "wetter" or "longer dry periods". I wonder if this is really a good basis to develop management plans for a catchment. Other short-time effects such as flood events or extreme precipitation events cannot be considered at a monthly scale. Consequently, it is impossible to discuss implications of those events on agriculture

or humans even if they might be more relevant than the general tendencies that are presented. Leaving out daily resolution might be also the reason for more or less similar tendencies for all climate scenarios. Based on the previous points, I do not share the opinion that this study provides a platform for further studies since high uncertainties are given due to methodical limitations (monthly resolution, exclusion of additional model output for further analyses).

In the following there are some minor comments:

L.34: I wonder if there are no recent studies that underline these statements. Examples are:

Gloria Salmoral, Bárbara A. Willaarts, Alberto Garrido, Björn Guse, Fostering integrated land and water management approaches: Evaluating the water footprint of a Mediterranean basin under different agricultural land use scenarios, In Land Use Policy, Volume 61, 2017, Pages 24-39, ISSN 0264-8377, https://doi.org/10.1016/j.landusepol.2016.09.027.

Pablo A. Mendoza, Naoki Mizukami, Kyoko Ikeda, Martyn P. Clark, Ethan D. Gutmann, Jeffrey R. Arnold, Levi D. Brekke, Balaji Rajagopalan, Effects of different regional climate model resolution and forcing scales on projected hydrologic changes, In Journal of Hydrology, Volume 541, Part B, 2016, Pages 1003-1019, ISSN 0022-1694, https://doi.org/10.1016/j.jhydrol.2016.08.010. Hartwich, J., Schmidt, M., Bölscher, J. et al. Environ Earth Sci (2016) 75: 1071. https://doi.org/10.1007/s12665-016-5870-4 I am pretty sure that there are many other studies that may be cited here.

L.49: I would use K as unit.

L.174: It seems very vague to me, if "Temporal Dynamics" is the right term for monthly and annual discharge.

L.180: Does this finding indicate that a daily resolution is appropriate to reduce model uncertainty?

[Figure]

L.186: It would be interesting to see if there are additional negative impacts on agriculture (e.g. higher floods, extreme precipitation) that limit the agricultural productivity. Since this analysis is based on monthly resolution, this aspect cannot be considered and consequently, the results are limited to more general statements (e.g. length of wet season).

L.259-269: This part is not a discussion but a summary of the results.

L.287: Of course it is of advantage to consider additional aspects beyond the outlet streamflow to discuss climate change and land use change impacts. However, the authors left out other approaches such as having a look at model output (streamflow components, water balance).

L.294: I do not share the opinion that this study provides a platform for further studies since high uncertainties are given due to methodical limitations (monthly resolution, exclusion of additional model output for further analyses).

L.329: In this regard I am not with the authors since model performance was evaluated with only a small number of performance measures and only for monthly resolution at a single gauge. Consequently, I do not see a satisfactory calibration.

Comments on figures and tables:

Fig.4: very unsharp, needs to be integrated in a higher Resolution

Fig.5: Why do the authors scaled up to 3000 m$^3$/month? 1500 would be enough and would allow having a closer look at the discharge. Please be consistent with scale resolution (e.g. fig a,b). It should be mentioned that it is modelled discharge.

Fig.6: I do not understand the term "1 in 10 year high".

Fig.9: Information in the right upper part is already given in the caption

I hope that the authors understand the listed comments as a recommendation to rework and improve some fundamental points of their manuscript. Hopefully, this study will be

published after a major revision. I wish much success in this.

---

## Referee Comment (RC2) · HHG Savenije (Referee) · 6 Dec 2017

This paper presents a well-documented and well-performed scenario study of climate and land-use changes in a Ghanaian rainforest catchment. As a case study, this is interesting for our audience, although one can doubt the innovation of the approach itself. Although it is essentially a scenario study to support management decisions, it may be considered under the category of 'cutting edge case studies', in a catchment where such studies have not been done before.

Regarding the impact of land use on runoff, it would have been good if the authors had specified how land-use is connected to the parametrization of the hydrological

model. It is not completely clear to me how land-use scenarios were coupled to model parameters and why certain values of these parameters were selected. As a result, it is very hard to judge how realistic the model results are in response to changing land use. I think that Section 2.2 and Section 2.3.1 should provide more detail on how land use is parameterized in the hydrological model, and how the model reflects the different land use scenarios.

Minor comments:

In line 77, please use the correct units [mm/y]. the mere mention of the word 'annual' in the text is no excuse for using the wrong unit. Precipitation is a flux and not a stock.

In line 23-34: 'first ever and the most current information' is a bit overstated. I suggest to write 'necessary information' instead

Figure 1 is hardly readable. Please use colours to distinguish the different boundaries and the river. Also the graphs would become clearer if clearer colours were used. For instance, Figure 11 is difficult to read.

In the caption of Table 3 write mean annual precipitation (MAP) in full.

---

## Author Comment (AC1) · 24 Jan 2018

Major comment 1: The manuscript "Scenario-based impacts of land use and climate changes on the hydrology of a lowland rainforest catchment in Ghana, West Africa" by Aduah et al. deals about the separate and combined analysis of impacts due to

climate and luse use change using the ACRU model. For this, a calibrated model for the Bonsa catchment was used (Aduah et al., 2017, companion paper). In general, impact studies for African catchments are of broad interest since adaptations plans regarding water management for the future will be necessary in the context of climate change. Hydrological or ecohydrological models can be a useful tool to support management decisions. However, the usefullness of model results strongly depends on a reasonable application of the models and a thorough analysis of the model results. In this regard I have two main concerns that need to be addressed or clarified by the authors:

1. All presented results are based on modelled monthly streamflow. Model calibration was presented in a companion paper. It was reported that "validation based on the daily time step did not generate satisfactory performance as NSE of 0.14 and 0.31 were obtained during calibration and validation, respectively". As a consequence monthly time steps were used since performance was better.

In my opinion, it is not good modelling practice to leave out poor model performance on a daily time step and to present satisfying model performance on a monthly basis. There must be a reason for poor model performance and in this regard, the authors need to clarify, if this poor performance may have implications for all following applications and conclusions. How can the authors be sure that hydrological processes are adequately simulated? Obviously there are not well simulated. Otherwise, model performance for daily time steps were much better.

Additionally it is not good modelling practice to use only a small number of performance measures, especially if all selected measures are focused on peak and high flow. I respect the circumstance that the investigated catchment may be not intensively monitored and that data scarcity may be a problem. However, there are additional ways to make sure that the model behaves reasonable and of course realistically (e.g. using constraints, rules-of-thumb, multi-site calibration, model output such as discharge components or hydrological components over time).

[Figure]

Response 1: The ACRU modelling was done at the daily time step and it would have been preferable to be able to undertake the analysis also at a daily time-step. However, the high levels of data uncertainty (limited rainfall, temp, gaps in streamflow spatially and temporally, etc) and constraints in the downscaling method applied to GCM data and the associated uncertainties in the Bonsa catchment (as discussed in Aduah et al, 2017) meant that a more acceptable time step for analysis was the monthly scale. Moriasi et al. (2007) show that if the Nash-Sutcliffe efficiency (NSE) index of between 0 and 1 indicates that model performance is generally acceptable, whereas a NSE of 0.5 and above indicates satisfactory model performance, whether at the monthly or daily time step. Thus, the NSE of 0.1 and 0.3 for the daily time step in the companion study does not necessarily indicate that the model performance in the Bonsa catchment was unacceptable; only that the model performance when analyzed at the monthly time step was better. Croke et al. (2008) highlight that higher resolution time steps, have the potential to have increased uncertainties. Thus, given the uncertainties in the input data due to the data scarcity in the catchment, analysis of modelling results at a monthly time step was deemed acceptable as it reduced uncertainty. It must be noted that the NSE was not the only performance measure used; several others, including differences between the means, standard deviations and graphical analysis of the time series (Aduah et al., 2017) which are not focused on the peak and high flows, were used and the current manuscript has been modified to include percent bias (PBIAS) statistic at the daily time-step, for model evaluation. However, multi-site calibration was not possible due to the lack of data, however, a sensitivity analysis was undertaken in the companion paper (Aduah et al., 2017) to assist in ensuring that the simulation was realistic. As the uncertainties in the poor data are reduced at a coarser time step, the authors believe that the monthly analysis of the modelling results is acceptable.

References Aduah, M. S., Jewitt, G. P. W., and Toucher, M. L. W.: Assessing suitability of the ACRU hydrological model in a rainforest catchment in Ghana, West Africa, Water Science, https://doi.org/10.1016/j.wsj.2017.06.001.
Moriasi, D. N., Arnold, J. G., Van Liew, M. W., Bingner, R. L., Harmel, R. D., and Veith, T. L.: Model evaluation guidelines for systematic quantification of accuracy in watershed simulations, Transactions of the Asabe, 50, 885-900, 2007.

Croke, B. F. W., Wagener, T., Post, D. A., Freer, J., and Littlewood, I.: Evaluating the information content of data for uncertainty reduction in hydrological modelling, 9th International Congress on Environmental Modelling and Software, 2008.

Major Comment 2: Due to coarse temporal resolution, all derived conclusions do have a more general character such as "wetter" or "longer dry periods". I wonder if this is really a good basis to develop management plans for a catchment. Other short-time effects such as flood events or extreme precipitation events cannot be considered at a monthly scale. Consequently, it is impossible to discuss implications of those events on agriculture or humans even if they might be more relevant than the general tendencies that are presented. Leaving out daily resolution might be also the reason for more or less similar tendencies for all climate scenarios. Based on the previous points, I do not share the opinion that this study provides a platform for further studies since high uncertainties are given due to methodical limitations (monthly resolution, exclusion of additional model output for further analyses).

Response 2: (Line 330 in revised manuscript). In the absence of any studies, we believe the information provided in this study will be useful to planners. We agree that floods/extremes will be useful, but given the high levels of uncertainty, we are reluctant to provide information of this nature at this stage. Hence, the longer time scales and more general statements were accepted as what could be offered by this study. We follow this approach because we are reluctant to suggest that current daily time-step output are useful for management/planning because of the uncertainties that exist now, but that monthly time-step analysis of the modelling results provide initial useful information which can be further unpacked in future at a daily time step, as improvements in driver data (e.g. satellite, improved monitoring etc) and in output from GCM downscaling, becomes available. Furthermore, the change factor method

of downscaling GCM climate projections has limitations which mean that it could only be applied at a monthly time scale. The conclusions reached in this study are therefore first statements about potential impacts of climate change and land use changes on hydrology and are thus useful since this is the first study on combined climate and land use change in the rainforest region of West Africa.

In the following there are some minor comments:

Comments: L.34: I wonder if there are no recent studies that underline these statements. Examples are: Gloria Salmoral, Bárbara A. Willaarts, Alberto Garrido, Björn Guse, Fostering integrated land and water management approaches: Evaluating the water footprint of a Mediterranean basin under different agricultural land use scenarios, In Land Use Policy, Volume 61, 2017, Pages 24-39, ISSN 0264-8377, https://doi.org/10.1016/j.landusepol.2016.09.027.

Pablo A. Mendoza, Naoki Mizukami, Kyoko Ikeda, Martyn P. Clark, Ethan D. Gutmann, Jeffrey R. Arnold, Levi D. Brekke, Balaji Rajagopalan, Effects of different regional climate model resolution and forcing scales on projected hydrologic changes, In Journal of Hydrology, Volume 541, Part B, 2016, Pages 1003-1019, ISSN 0022-1694, https://doi.org/10.1016/j.jhydrol.2016.08.010. Hartwich, J., Schmidt, M., Bölscher, J. et al. Environ Earth Sci (2016) 75: 1071. https://doi.org/10.1007/s12665-016-5870-4

I am pretty sure that there are many other studies that may be cited here.

Response: Comments are accepted. More recent references have been included in the introduction section.

References added Guzha, A. C., Rufino, M. C., Okoth, S., Jacobs, S., and Nóbrega, R. L. B.: Impacts of land use and land cover change on surface runoff, discharge and low flows: Evidence from East Africa, Journal of Hydrology: Regional Studies, 15, 49-67, https://doi.org/10.1016/j.ejrh.2017.11.005, 2018.

Mwangi, H. M., Julich, S., Patil, S. D., McDonald, M. A., and Feger, K.-H.: Relative contribution of land use change and climate variability on discharge of upper Mara River, Kenya, Journal of Hydrology: Regional Studies, 5, 244-260, https://doi.org/10.1016/j.ejrh.2015.12.059, 2016.

Veettil, A. V., and Mishra, A. K.: Water security assessment using blue and green water footprint concepts, Journal of Hydrology, 542, 589-602, https://doi.org/10.1016/j.jhydrol.2016.09.032, 2016.

Comments: L.49: I would use K as unit. Response: The degree unit (line 51 of revised manuscript) has been changed to K.

Comments: L.174: It seems very vague to me, if "Temporal Dynamics" is the right term for monthly and annual discharge.

Response: The comment is accepted. Sections 3.1.1 and 3.2.1 headings have been changed to "Temporal Patterns". The sections show annual and seasonal differences over time, hence the description as temporal patterns.

Comment: L.180: Does this finding indicate that a daily resolution is appropriate to reduce model uncertainty?

Response: Daily simulation was done, but due to uncertainties, the analysis of the results was done at the monthly time-step (line 220 in revised manuscript).

Comments: L.186: It would be interesting to see if there are additional negative impacts on agriculture (e.g. higher floods, extreme precipitation) that limit the agricultural productivity. Since this analysis is based on monthly resolution, this aspect cannot be considered and consequently, the results are limited to more general statements (e.g. length of wet season).

Response: Yes, this is possible with the model, but confidence in the downscaled climate data to daily time-step (Kusangaya et al., 2017) means we chose not to do this analysis. However, the more general statements on the impacts on agricultural productivity are still useful as the rain-fed agriculture in Ghana is sensitive to changes

in the wet season as highlighted by, e.g. Anim-Kwapong and Frimpong (2005) for cocoa and Jalloh et al. (2013) for crops in West Africa (please see line 226 in revised manuscript).

References Anim-Kwapong, G. J., and Frimpong, E. B.: Vulnerability of agriculture to climate change- impact of climate change on cocoa production Cocoa research institute of Ghana, New Tafo Akim, Ghana, 2005.

Jalloh, A., Faye, M. D., Roy-Macauley, H., Sérémé, P., Zougmoré, R., Thomas, T. S., and Nelson, G. C.: Overview, International Food Policy Research Institute(IFPRI), Washington, DC, USA, 1-36, 2013.

Kusangaya, S., Warburton, M., and Archer van Garderen, E.: Use of ACRU, a distributed hydrological model, to evaluate how errors from downscaled rainfall are propagated in simulated runoff in uMngeni catchment, South Africa, Hydrological Sciences Journal, 62, 1995-2011, 10.1080/02626667.2017.1349317, 2017.

Comment L.259-269: This part is not a discussion but a summary of the results.

Response: The comments are accepted. The section has been modified to explain the results (see line 299-305 in revised manuscript).

Comments L.287: Of course it is of advantage to consider additional aspects beyond the outlet streamflow to discuss climate change and land use change impacts. However, the authors left out other approaches such as having a look at model output (streamflow components, water balance).

Response: Comments about line 315(in revised manuscript) are accepted. However, if the available data had allowed analysis of the modelling output at a finer time scale, it would have been useful to consider these, but due to the uncertainties created and the limitation of the change factor method of downscaling climate projections, it was not done. The manuscript has been modified to include these as recommendations (see line 365-367 in revised manuscript).

[Figure]

Comment L.294: I do not share the opinion that this study provides a platform for further studies since high uncertainties are given due to methodical limitations (monthly resolution, exclusion of additional model output for further analyses).

Response: The phrase "provides a platform for further studies" has been maintained in the manuscript (line 330, revised manuscript). This is because the ACRU model is capable of running at a daily time-step, but that analysis for this study was at a monthly resolution. Therefore as the data becomes available, the analysis can also take place at a daily time-step. Hence this study does provide a platform for further studies. Please refer to Response 2 on page 2 for further discussion on this.

Comments L.329: In this regard I am not with the authors since model performance was evaluated with only a small number of performance measures and only for monthly resolution at a single gauge. Consequently, I do not see a satisfactory calibration.

Response: (line 365-367 revised manuscript). Percent bias (PBIAS) at the daily time-step (-.3.8% and -15.6% for calibration and validation, respectively) has been included in the model evaluation in the manuscript, which further indicates that the modelling at the daily time-step was not unacceptable, only that the analysis at the monthly resolution produced better statistics. Please refer to Response 1 on page 1, which addressed the Major comment 1 for a further discussion on this.

Comments on figures and tables:

Comment: Fig.4: very unsharp, needs to be integrated in a higher Resolution Response: Resolution of Figure 4 has been improved.

Comment: Fig.5: Why do the authors scaled up to 3000 m3/month? 1500 would be enough and would allow having a closer look at the discharge. Please be consistent with scale resolution (e.g. fig a,b). It should be mentioned that it is modelled discharge.

Response: Scaling of Figure 5 has been reduced to maximum 1500 m3/month

Comment: Fig.6: I do not understand the term "1 in 10 year high".

[Figure]

Response: To aid in clarity, both the terms "1 in 10 year high" and "1 in 10 year low" have been removed from Figure 6.

Comment: Fig.9: Information in the right upper part is already given in the caption

Response: The caption of Figure 9 has been corrected. 90th and 10th percentile have been removed from the figure caption.

Comment: I hope that the authors understand the listed comments as a recommendation to rework and improve some fundamental points of their manuscript. Hopefully, this study will be published after a major revision. I wish much success in this.

Response: The authors thank the reviewer for his/her comments. Each of the comments made has been given considered thought and responded to. We hope that the clarifications made and the justifications given address the reviewer's comments adequately.

---

## Author Comment (AC2) · 24 Jan 2018

Responses to Referee# 2 Comment: This paper presents a well-documented and well-performed scenario study of climate and land-use changes in a Ghanaian rainforest catchment. As a case study, this is interesting for our audience, although one can doubt the innovation of the approach itself. Although it is essentially a scenario study to support management decisions, it may be considered under the category of 'cutting edge case studies', in a catchment where such studies have not been done before.

Regarding the impact of land use on runoff, it would have been good if the authors had specified how land-use is connected to the parametrization of the hydrological

model. It is not completely clear to me how land-use scenarios were coupled to model parameters and why certain values of these parameters were selected. As a result, it is very hard to judge how realistic the model results are in response to changing land use. I think that Section 2.2 and Section 2.3.1 should provide more detail on how land use is parameterized in the hydrological model, and how the model reflects the different land use scenarios.

Response for section 2.2 Section 2.2 has been improved with details of the land use parametrization used by the ACRU hydrological model.

The improvement (lines 91-109 in revised manuscript) reads as follows: "The physically-based conceptualisation of the land cover characteristics and its various interactions with the hydrological processes in the ACRU model, means that the structure of the model demonstrates a high sensitivity to changes in land cover, land use and land management (Schulze et al., 1995; Warburton et al., 2010). Therefore, the ACRU model is able to simulate the impacts of land cover and land use change on water flows. The land cover is conceptualized in ACRU by using vegetation and water use input parameters that describe the land use processes and how the hydrological processes are governed by the vegetation. The above-ground vegetation properties in the ACRU model are conceptualised through the consumptive water use of vegetation which is expressed as a monthly crop coefficient (ACRU variable name = CAY) and canopy interception loss as either a monthly canopy interception losses per rain-day (VEGINT) or calculated from the monthly Leaf Area Index (LAI). The rainfall abstracted by interception, surface detention storage and initial infiltration before stormflow commences is represented in the ACRU model through the coefficient of initial abstraction (COIAM). The soil water content of the A-horizon, infiltrability and initial abstractions influence the daily soil water budget (Schulze, 1995). The presence and amount of litter and/or mulch, which has the potential to reduce and/or prevent soil erosion and soil water evaporation losses (Schulze, 2007), is accounted for through the required input of the percentage surface cover (PCSUCO). The below ground processes are concep-

[Figure]

tualised through three root related parameters and a plant soil water stress indicator. The three root parameters required are the total depth of the root profile (EFRDEP), the percentage of active roots in the A-horizon (ROOTA) and lastly the degree of root colonisation in the soil horizons (COLON). To account for the onset of plant stress, the fraction of plant available water in the soil horizons at which total evaporation is assumed to drop below maximum evaporation due to drying of the soil needs to be input."

Response for section 2.3.1 Section 2.3.1 has also been improved with details of the land use information used in simulating impacts on streamflow changes in the Bonsa catchmemt.

The improvement (lines 138-147 in revised manuscript) reads as follows: The land use scenarios were obtained from (Aduah et al., under review). For each subcatchment the area of the land uses present in the baseline, current and future scenarios were determined, and used to create the HRU's. Thus, the areas of the various HRU's within the subcatchments varied between the different scenarios. For each land use type the parameters for the vegetation and water use variables described above were determined. The initial values of the ACRU model parameters were adopted from South African (Schulze, 1995;Warburton et al., 2012) and Ghanaian (Bekoe, 2005) case studies for a sensitivity and calibration study in the Bonsa catchment (Aduah et al., 2017), after which the final parameter values were selected. The original parameters of the ACRU model were derived by Schulze (1995) based on a working rule that linked the parameters to mean annual precipitation, monthly heat units, soil water status and crop physiology (Warburton et al., 2012). During the sensitivity analysis and calibration study the PCSUCO for all the land uses were estimated, based on field observations.

Minor comment #1: In line 77, please use the correct units [mm/y]. the mere mention of the word 'annual' in the text is no excuse for using the wrong unit. Precipitation is a flux and not a stock.

Response: Units (in line 80 of revised manuscript) have been changed to mm/year

Minor comment #2: In line 23-34: 'first ever and the most current information' is a bit overstated. I suggest to write 'necessary information' instead

Response: 'first ever and the most current information' (line 25-26 of revised manuscript) has been changed to 'necessary information'.

Minor comment #3: Figure 1 is hardly readable. Please use colours to distinguish the different boundaries and the river. Also the graphs would become clearer if clearer colours were used. For instance, Figure 11 is difficult to read.

Response: Figure 1 has been redrawn with colour and thickness of some of the lines have been increased to improve contrast and legibility.

Figure 11 as well as all the other figures have been improved with clearer colours and thicker lines.

Minor comment #4: In the caption of Table 3 write mean annual precipitation (MAP) in full. Response: In Table 3, MAP has been written in full as mean annual precipitation.